# Pain and Motor Function in Myotonic Dystrophy Type 1: A Cross-Sectional Study

**DOI:** 10.3390/ijerph20075244

**Published:** 2023-03-23

**Authors:** Sara Liguori, Antimo Moretti, Giuseppe Toro, Marco Paoletta, Angela Palomba, Giuseppe Barra, Francesca Gimigliano, Giovanni Iolascon

**Affiliations:** 1Department of Medical and Surgical Specialties and Dentistry, University of Campania “Luigi Vanvitelli”, Via De Crecchio n. 4, 80138 Naples, Italy; sara.liguori@unicampania.it (S.L.);; 2Post Intensive Functional Rehabilitation Unit, Istituto di Diagnosi e Cura Hermitage Capodimonte, 80131 Naples, Italy; 3Department of Mental and Physical Health and Preventive Medicine, University of Campania “Luigi Vanvitelli”, Largo Madonna delle Grazie n. 1, 80138 Naples, Italy

**Keywords:** pain, myotonic dystrophy type 1, gender, fatigue, balance, gait, disability evaluation

## Abstract

Pain is an underestimated finding in myotonic dystrophy type 1 (DM1). We provide a characterization of pain in terms of functional implications through a multidimensional assessment in patients with DM1, focusing on gender differences. We assessed pain through the Brief Pain Inventory (BPI) and its indexes (the Severity Index (SI) and the Interference Index (II)), balance/gait (the Tinetti Performance-Oriented Mobility Assessment (POMA)), functional abilities (the Functional Independence Measure (FIM)), and fatigue (the Fatigue Severity Scale (FSS)). We divided our sample into a mild (<4) and a moderate–severe group (≥4) based on BPI indexes. A between-group analysis was performed. We recruited 23 males and 22 females with DM1. A statistically significant difference was found for the FSS and the BPI-SI ≥ 4, and for all outcomes in the BPI-II ≥ 4 (*p* ≤ 0.003). In the female group, all outcomes except for the FIM were statistically significantly worse (*p* ≤ 0.004). Dividing our sample into four groups based on gender and the BPI, a statistically significant difference was found for FSS between the two groups with BPI-II ≥ 4 (with worsen score in the female one) (*p* < 0.002). Pain in DM1 patients is highly reported and gender related, with increased fatigue and poor balance/gait in the female group.

## 1. Introduction

Myotonic dystrophy type 1 (DM1), also known as Steinert’s disease, is the most common adult-onset muscular dystrophy [1]. The prevalence is approximately 1 in 8000 (12.5 per 100,000) but it is probably underestimated due to the difficulty in identifying patients affected [2]. DM1 is an autosomal dominant disease associated with an abnormal repetition of the cytosine-thymine-guanine (CTG) triplet in the 3′ untranslated region of the protein kinase DM1 (DMPK) gene on chromosome 19q13.3 [3]. DM1 can be divided into different clinical phenotypes based on the age of onset of the first symptoms [4,5]. Classic DM1 generally occurs around the second–fourth decade and is characterized by muscle weakness and wasting, myotonia, cataract, and cardiac conduction abnormalities [6]. When DM1 begins in infancy, cognitive and learning impairments are more frequent than muscular ones [7]. Based on the CTG repeat sizes, typically four phenotypes are described: (1) 50 to 150 CTG repeats correspond to a mild phenotype; (2) 50 to 1000 CTG repeats a “classic” phenotype; (3) >800 CTG repeats an “early onset” phenotype, and (4) >1000 CTG repeats a “congenital” phenotype [8].

As in other neuromuscular disorders (NMD) [9,10], pain is considered a common symptom in affected people, involving approximately 60% of individuals with DM1 [11], and it seems related with vital capacity, illness duration, CTG expansion, quality of life, and fatigue [12]. It was hypothesized that increased muscle susceptibility to contraction-induced injury during DM1 progression may contribute to musculoskeletal pain [11]. Furthermore, this population often reports painful muscle cramping due to the prolonged muscle contracture that characterizes myotonic phenomenon [13]. In this context, myopathy per se causes progressive muscle wasting and fatty infiltration that might be associated with nociceptive pain [14,15].

In DM1 patients, a significant contribution of the neuropathic mechanism as a pain generator has also been postulated, considering that peripheral neuropathy was reported in this population [16].

As typically observed in other NMDs, the chronicization of pain in patients with myotonic dystrophy may adversely affect quality of life and social participation, with increased impact on the health care system [17,18]. Finally, it was documented that pain in DM1 is influenced by gender, with increased pain sensitivity in females [12].

However, despite the huge disabling and socio-economic impact of pain in NMDs, this impairment is often overlooked in DM1 patients. Moreover, pain in DM1 has not yet been investigated from a multidimensional perspective, and less is known about its potential association with some disabling aspects of the disease such as gait and balance disturbances. The aim of this study is to provide a better characterization of pain in terms of both intensity and functional implications through a multidimensional assessment in a cohort of patients with DM1, particularly focusing on gender differences.

## 2. Materials and Methods

### 2.1. Participants

In our observational, cross-sectional study, we recruited patients with a genetic diagnosis of DM1, classic phenotype, referred to our outpatient rehabilitation service. All patients provided written informed consent, in accordance with the guidelines of the Declaration of Helsinki. The Ethical Committee of the Università della Campania “Luigi Vanvitelli” approved the study protocol (Committee’s reference number: 0017390/2020).

### 2.2. Outcomes

All patients were assessed by a dedicated evaluation protocol, including the collection of anamnestic and anthropometric data, and the administration of the following outcome measures:

#### 2.2.1. The Brief Pain Inventory (BPI)

The Brief Pain Inventory is a self-reported questionnaire used to evaluate pain intensity and its interference with activities of daily living (ADLs) with two indexes (the Severity Index (SI) and the Interference Index (II)) [19]. The BPI-SI includes four items (worst, least, average pain in the last 24 h and current pain intensity) with a score ranging from 0 (absence of pain) to 10 (worst possible pain) for each item and a total BPI-SI score using the mean score of the four items. The total possible score ranges from 0 to 10, where 1–4 indicates “mild pain”, 5–6 “moderate pain”, and 7–10 “severe pain” [20]. The BPI-II investigates the interference of pain on seven items (general activity, mood, walking ability, normal work, relations with other people, sleep, and enjoyment of life) with a sub-item score ranging from 1 (no interference) to 10 (total interference); the total score is determined from the mean of the BPI-II items [21].

According to the World Health Organization (WHO) classification of pain [16,22], we divided our population into a mild (1–3.9) and a moderate–severe group (4–10), based on the BPI-SI (Group 1-SI and 2-SI) and the BPI-II (Group 1-II and 2-II).

#### 2.2.2. The Tinetti Performance-Oriented Mobility Assessment (POMA)

The Tinetti Performance-Oriented Mobility Assessment is a clinical test used to evaluate gait and balance. The balance subscale includes 9 items and a total score ranging from 0 to 16. The gait subscale has 7 items, with a total score ranging from 0 to 12. A lower total POMA score (<15.5) corresponds to a greater risk of fall [23].

#### 2.2.3. The Functional Independence Measure (FIM)

The FIM is an 18-item ordinal scale that evaluates functional abilities in six areas—self-care, continence, mobility, transfers, communication, and cognition. For each item, it is possible to give a score ranging from 1 to 7 based on the level of independence (1 = requiring total assistance, 7 = completely independent). The total score will be a value between 18 and 126 and the cut-off for a poor outcome is considered 63 [24].

#### 2.2.4. The Fatigue Severity Scale (FSS)

The Fatigue Severity Scale is a 9-item questionnaire aiming to assess the severity of fatigue symptoms and their impact on daily functioning. Each question is scored from 1 (“completely disagree”) to 7 (“completely agree”). The total score is given by the mean score of the 9 items. A total score >36 suggests a high level of fatigue [25].

In order to investigate the role of gender on symptoms and functional outcomes, we also analyzed any difference between male and female groups according to pain severity and interference.

### 2.3. Statistical Analysis

Statistical analysis was carried out using the Statistical Package for the Social Sciences 25 (SPSS 25; IBM Corp., Armonk, NY, USA) software. Data for continuous variables are expressed as the means ± standard deviations or as the median (interquartile range (IQR)). Categorical data are reported in terms of counts (absolute numbers and percentage). Normality was checked through the Shapiro–Wilk test. We performed a multivariate logistic regression analysis to measure if pain severity and interference are influenced by disease duration (DD), FSS, FIM and POMA scores. We evaluated between-group differences for all the outcome measures. The independent-samples Mann–Whitney U test was used to compare POMA, FSS and FIM scores among the groups and for the sub-analysis on gender (male-female). Effect size (η^2^) was reported, indicating a small (0.01), medium (0.06) or large (0.14) effect [26]. Finally, we divided our sample into 4 groups based on gender (male/female) and the BPI-SI (<4/≥4) and 4 groups based on gender (male/female) and the BPI-II (<4/≥4), to compare POMA, FSS, and FIM scores among the groups. The independent-samples Kruskal–Wallis test was used for this comparison; significance values have been adjusted by the Bonferroni correction for multiple tests; effect size (ε^2^) was reported, indicating 0.00 < 0.01—negligible, 0.01 < 0.04—weak, 0.04 < 0.16—moderate, 0.16 < 0.36—relatively strong, 0.36 < 0.64—strong and 0.64 < 1.00—very strong effect [27]. We considered a significance threshold of *p* < 0.05.

## 3. Results

A total of 45 DM1 patients (23 males and 22 females) were recruited. In Table 1, we reported the baseline characteristics of the population. According to pain measures, 9 patients experienced absence of pain (20%), 11 mild pain (24.44%) and 25 moderate–severe pain (55.56%); moreover, 9 patients reported an absence of interference in the ADLs (20%), 12 patients (26.66%) a minimal pain interference, and 24 moderate–severe interference in the ADLs (53.34%) (Figure 1). The frequency of painful areas in our population are shown in Figure 2. Table 2 showed that both fatigue and BMI influence the severity of pain and fatigue also influences the interference of pain with ADLs. Based on the BPI-SI, a statistically significant difference was found between groups for the fatigue outcome, while for the BPI-II, all outcomes showed a statistically significant difference among groups (Figure 3).

Figure 4 reported the changes between gender in the BPI, POMA, and FSS scores, which were statistically significantly worse in the female DM1 group.

Finally, dividing our sample into four groups based on gender and BPI sub-indexes:-For the BPI-SI (Figure 5a–c), a statistically significant difference was found for the FSS with a lower score in male group with less severe pain compared to the female group with more severe pain (*p* < 0.002);-For the BPI-II (Figure 5d–f), a statistically significant difference was found for the FSS between the two groups with BPI-II ≥ 4 (with worsen score in the female one) (*p* < 0.002) and for the FSS and the POMA, with a lower score in the male group with the BPI-II ≤ 4 compared to the female group with BPI-II ≥ 4 (*p* < 0.002).

## 4. Discussion

Neuromuscular disorders are characterized by several functioning impairments, including pain, although this symptom is often underdiagnosed and undertreated. In this scenario, we performed a cross-sectional study aiming to fill the gap about the knowledge on pain patterns in DM1 population in terms of intensity and functional implications, also focusing on gender differences.

### 4.1. A Multidimensional Assessment of Pain for People with DM1

In our cohort, 80% of patients with DM1 reported musculoskeletal pain with low back region mostly involved (55.6%). This finding suggests that the prevalence of low back pain in our population is higher than that reported in age-matched individuals, who have a prevalence of low back pain ranging from 28% to 42% [28]. Our anecdotal experience suggests that the impact of pain on DM1 patients might be even more significant. Therefore, to better investigate the impact of pain on key functional outcomes of DM1 patients, we administered the original version of the BPI, which addresses pain intensity and interference with ADLs in the last 24 h, whereas in a previous study, Peric et al. reported a higher frequency of patients with pain (88.5%), with a mean BPI-SI score of 4.6 ± 2.3 using a modified version of the BPI on 52 participants asking about the pain experienced in the last four weeks [10]. These findings are consistent with the results in our population only in terms of the mean BPI-SI [4.40 (0–8)].

Multidimensional scales such as the BPI, although complex to administer, seem to demonstrate good reliability and validity for many clinical situations. Moreover, the BPI is cross-cultural as it was validated in several languages, including Italian. However, the most used assessment tools for pain are the unidimensional scales (e.g., Numerical Rating Scale (NRS) and the Visual Analogue Scale (VAS)). These tools, although easy to understand and quicker to administer, assign a single number to the pain experience and do not have psychometric properties that may discriminate among equivalent intervals in terms of scaling the intensity of pain (ceiling effect) [12]. Solbakken et al. addressed pain severity in DM1 individuals using the NRS, reporting a global mean score of 4.6 ± 2.7 [12]. A similar mean NRS score was reported by Miro et al. (4.5 ± 2.6), although the authors also included facioscapulohumeral muscular dystrophy (FSHD) patients [9].

According to pain distribution, our study confirms that the most involved anatomic region is the lower back (55.6%) [9,10,11,12]. These data may be justified by the progressive wasting of trunk muscles in DM1 patients, in particular of the lower erector spinae muscles, which may promote a muscular imbalance and persistent mechanical damage, resulting in pain onset and persistence [12]. Otherwise, the main disease feature itself (i.e., myotonia) induces muscular stiffness that probably plays a role as a pain generator.

For pain interference in ADLs, our patients reported a low BPI-II in 26.66% of cases (score 1–3.9), and a high BPI-II (score 4–10) in 53.34% of cases. Miro et al. conducted a similar study using a modified version of the BPI-II, adapting the items to consider the unique characteristics of the study population [9]. The authors reported a mean score of 3.22 ± 2.53 of the BPI-II. Differently from our study, in which most of the population reported a high interference of DM1 pain with ADLs, in the study of Miro et al., the maximum score never exceeded 5 (low–moderate interference with ADLs). These controversial findings can be explained considering that the authors also included FSHD patients in their study, thus requiring caution when generalizing the results.

### 4.2. Pain and Fatigue in People with DM1

As for fatigue, our participants reported a median FSS score of 43 (9–63), suggesting a moderate perceived fatigue. Moreover, fatigue seems to influence both pain severity and interference with ADLs and the FSS score was higher in individuals reporting a BPI-SI and -II ≥4, compared to those with lower scores (<4), with a large effect size (η^2^ ≥ 0.1). This relationship between pain and fatigue has also been reported by previous studies on DM1 patients [10,12], where authors suggested pain as a putative predictor of fatigue onset, particularly in females.

### 4.3. Pain and Gait/Balance in People with DM1

The POMA is a mainstay in the evaluation of balance and gait in different populations and could be considered as a surrogate outcome for muscle function and fall risk assessment. In our study, we found an association between poor POMA scores and high BPI-II in the DM1 population, with a large effect size (η^2^ ≥ 0.1). In our opinion, pain could adversely affect muscle function in DM1 patients, also resulting in the “reflex muscle inhibition” that might slow neuromuscular response, thus contributing to an increased risk of falls.

### 4.4. Pain and Functional Abilities in People with DM1

Concerning functional independence in the DM1 population, in our study, we decided to use the FIM that showed a worse mean score in individuals with a higher BPI-II. Landfeldt et al., in a cross-sectional study on 152 DM1 patients, reported high levels of disability assessed through the DM1-ActivC mainly attributable to lower muscle strength and poor coordination, underestimating a potential role of musculoskeletal pain [29]. Conversely, in line with our research, Solbakken et al. suggested a correlation between pain and performance in ADLs in DM1 patients [12]. We can speculate that interference of pain with ADLs might be a consequence of a competition between pain and cognitive tasks for recruiting attentive resources. Generally, engaging in a distractive task might reduce pain perception and carefully performing a task induces an analgesic effect, even if individuals with chronic pain might experience impaired executive function and reduced attentional resources [30]. These concepts might partially explain our findings about the relationship between pain perception and disability. This association is strengthened in our female population, although the mechanisms underlying gender disparity have not yet been clarified, probably due to complex interactions between biological, functional, and socio-cultural factors.

### 4.5. Pain, Functional Outcomes and Gender in People with DM1

According to a previous study [12], in our cohort, we reported a significantly worse score for the BPI, the POMA, and the FSS in the female DM1 group.

These gender-related findings seem to have a biological basis, as demonstrated by a murine model describing the synergic role of Acid-Sensing Ion Channel Subunit 3 (ASIC3; a sensor of acidic pain and integrators of molecular signals produced during inflammation) with testosterone to protect against muscle fatigue. Gender differences about fatigue onset might depend on both the presence of testosterone and the activation of the ASIC3 protein [31]. Consistent with these findings, we found significant gender differences for pain, fatigue, and balance/gait, with worse scores in our female sample in all these outcomes. In particular, Solbakken et al. hypothesized a possible relationship between pain and activity limitations in DM1 women, which might be explained by lower muscle strength in women than in men according to the higher BPI scores reported in females [12]. However, the sub-analysis stratified for gender and the BPI showed that higher pain interference was associated with a higher level of perceived fatigue in both sexes, with a worse score in the pairwise comparison for the female group, with a moderate effect size (ε^2^ = 0.30).

### 4.6. Limitations of Our Study

Our study presents several limitations: (1) the cross-sectional design does not enable establishing a cause–effect relationship since the outcome and exposure are evaluated at the same time; (2) the sample of DM1 patients included is too small to draw strong conclusions, even if DM1 is a rare disease and this makes our population quite consistent; (3) the absence of a healthy control group does not permit establishing whether the observed events are specific for the disease; (4) the neuropathic component of pain was not addressed in our population, although peripheral neuropathy and conduction disturbance along the dorsal column–medial lemniscus pathway (i.e., the spinal somatosensory tract) was found in this condition. In the future, we will aim to address this issue in our cohort using the recently validated Italian version of the Leeds Assessment of Neuropathic Symptoms and Signs Scale and the Pain DETECT Questionnaire [32].

## 5. Conclusions

DM1 is a rare disease with several clinical manifestations, including musculoskeletal pain as a common but underestimated finding. In our population, four out of five DM1 patients reported musculoskeletal pain, mostly in the lower back. In our cohort, our data suggest that pain perception in DM1 patients is gender related and has significant functional implications on clinical practice in terms of increased fatigue and poor balance and gait, despite no relevant difference in functional independence compared to male patients.

In our opinion, pain assessment and treatment should be routinely addressed during the comprehensive management of people affected by DM1.

## Figures and Tables

**Figure 1 ijerph-20-05244-f001:**
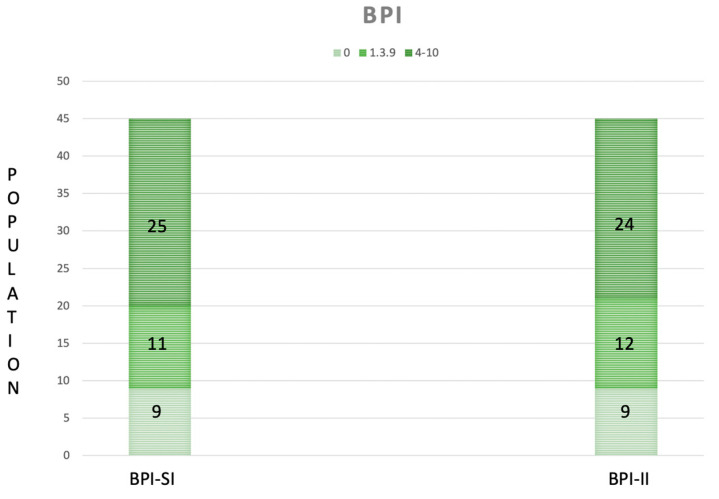
The Severity Index (SI) and the Interference Index (II) of the Brief Pain Inventory (BPI), categorized according to the World Health Organization as absence (0), mild (1–3.9) and moderate–severe pain (4–10) in our population (n = 45). The green ranges from being light in the case of the absence of pain severity/interference to being dark in the case of moderate to severe pain severity/interference.

**Figure 2 ijerph-20-05244-f002:**
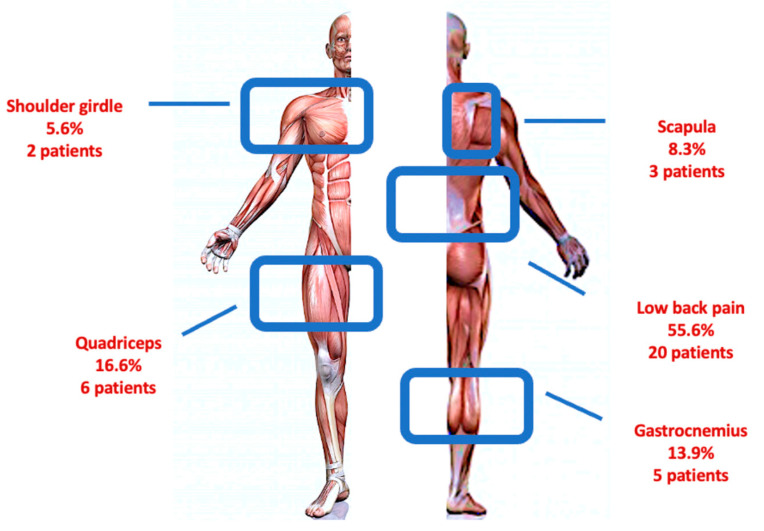
Painful anatomical region distribution in our cohort of patients with DM1.

**Figure 3 ijerph-20-05244-f003:**
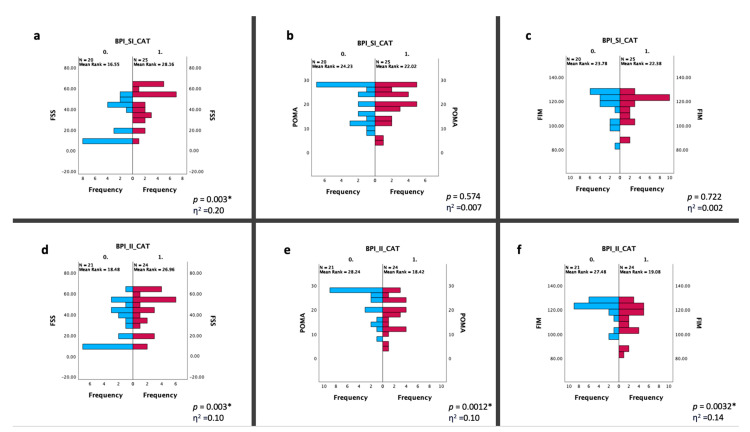
Statistical changes between two groups based on the BPI-SI (**a**–**c**) and the BPI-II (**d**–**f**). Statistical analysis was performed using the independent-samples Mann–Whitney U test (* *p* < 0.05). Effect size (η^2^) reported as small (0.01), medium (0.06) or large (0.14). Graphs (**a**–**c**) show the differences in the distribution of frequency between the BPI-SI CAT 0 (BPI-SI < 4-blue bars) and 1 (BPI-SI ≥ 4-red bars) in terms of the FSS, POMA and FIM scores; similarly, the graphs (**d**–**f**) show the differences in the distribution of frequency between the BPI-II CAT 0 (BPI-II < 4-blue bars) and 1 (BPI-II ≥ 4-red bars) about the same outcomes. Abbreviations: BPI-SI: Brief Pain Inventory Severity Index; BPI-II: Brief Pain Inventory Interference Index; CAT: category; FSS: Fatigue Severity Scale; POMA: Tinetti Performance-Oriented Mobility Assessment; FIM: Functional Independence Measure.

**Figure 4 ijerph-20-05244-f004:**
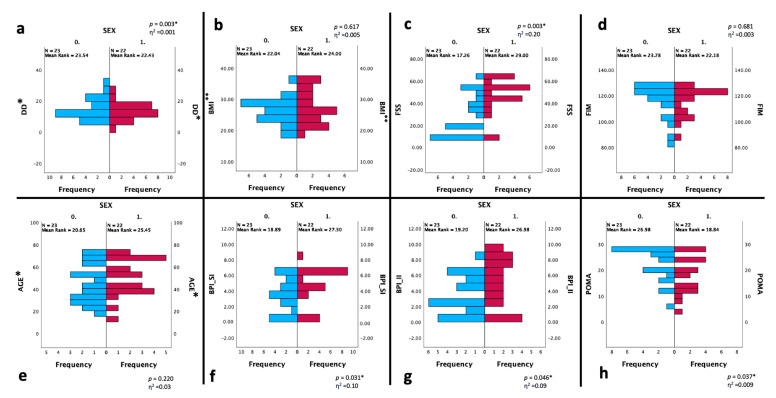
Statistical changes between two groups based on the gender (n = 45). Statistical analysis was performed using the independent-samples Mann–Whitney U test (* *p* < 0.05). Effect size (η^2^) reported as small (0.01), medium (0.06) or large (0.14). Graphs show the differences in the distribution of frequency between SEX CAT 0 (male—blue bars) and 1 (female—red bars) about the DD (**a**), BMI (**b**), FSS (**c**), FIM (**d**), age (**e**), BPI-SI (**f**), BPI-II (**g**), POMA (**h**) scores. Abbreviations: CAT: category; DD: disease duration; BMI: Body Mass Index; BPI-SI: Brief Pain Inventory Severity Index; BPI-II: Brief Pain Inventory Interference Index; FSS: Fatigue Severity Scale; POMA: Tinetti Performance-Oriented Mobility Assessment; FIM: Functional Independence Measure. * expressed in years; ** expressed in kg/m^2^.

**Figure 5 ijerph-20-05244-f005:**
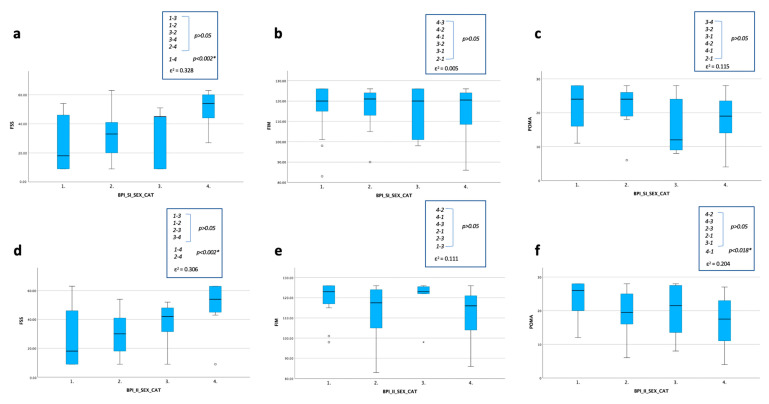
Statistical changes between four groups based on gender and the BPI-SI (**a**–**c**) and the BPI-II (**d**–**f**) (n = 45). Statistical analysis was performed using the independent-samples Kruskal–Wallis test; significance values have been adjusted by the Bonferroni correction for multiple tests (* *p* < 0.05). The effect size (ε^2^) reported as 0.00 < 0.01—negligible, 0.01 < 0.04—weak, 0.04 < 0.16—moderate, 0.16 < 0.36—relatively strong, 0.36 < 0.64—strong and 0.64 < 1.00—very strong. Graphs show the differences in the distribution of frequency in terms of the FSS, POMA and FIM scores among the 4 groups distinguished for the BPI-SI-SEX-CAT (**a**–**c**) and the BPI-II-SEX-CAT (**d**–**f**): 1 male with BPI sub-index < 4; 2 male with BPI sub-index ≥ 4; 3 female with BPI sub-index < 4; 4 female with BPI sub-index ≥ 4. Abbreviations: BPI-SI: Brief Pain Inventory Severity Index; BPI-II: Brief Pain Inventory Interference Index; CAT: category; FSS: Fatigue Severity Scale; POMA: Tinetti Performance-Oriented Mobility Assessment; FIM: Functional Independence Measure.

**Table 1 ijerph-20-05244-t001:** Baseline characteristics of our cohort of DM1 patients (N = 45).

Variable	DM1 Patients (N = 45)
Age (years)	47.33 ± 17.15
BMI (kg/m2)	26.84 ± 4.84
Gender Male (%) Female (%)	23 (51.11%) 22 (49.89%)
DD (years)	14.00 ± 5.85
BPI-SI	4.40 (0–8)
BPI-II	4.20 (0–9)
POMA	20 (4–28)
FIM	121 (83–126)
FSS	43 (9–63)

Continuous variables are expressed as the mean ± standard deviation or the median (IQR, interquartile range); discrete ones are expressed as the total number (%). Abbreviations: DD: disease duration; BMI: Body Mass Index; BPI-SI: Brief Pain Inventory Severity Index; BPI-II: Brief Pain Inventory Interference Index; POMA: Tinetti Performance-Oriented Mobility Assessment; FIM: Functional Independence Measure; FSS: Fatigue Severity Scale.

**Table 2 ijerph-20-05244-t002:** Multivariate logistic regression analysis to measure the influence of the DD, FSS, FIM and POMA scores on pain severity and interference with ADLs.

**HYPOTHESIS** **(Regression Weights)**	**Beta Coefficient**	***t*-Value**	***p*-Value**
DD- > BPI-SI	−0.093	−1.533	0.134
AGE- > BPI-SI	0.040	2.018	0.051
SEX- > BPI-SI	−0.515	−0.638	0.527
BMI- > BPI-SI	0.157	2.295	0.028 *
FSS- > BPI-SI	0.064	2.954	0.005 *
FIM- > BPI-SI	0.051	1.229	0.227
POMA- > BPI-SI	−0.032	−0.492	0.626
**HYPOTHESIS** **(Regression Weights)**	**Beta Coefficient**	***t*-Value**	***p*-Value**
DD- > BPI-II	−0.041	−0.553	0.583
AGE-> BPI-II	0.044	1.815	0.078
SEX-> BPI-II	−0.099	−0.100	0.921
BMI-> BPI-II	0.135	1.622	0.113
FSS-> BPI-II	0.060	2.285	0.028 *
FIM-> BPI-II	0.020	0.386	0.702
POMA-> BPI-II	−0.062	−0.779	0.441

Abbreviations: ADLs: activity of daily living; DD: disease duration; BMI: Body Mass Index; BPI-SI: Brief Pain Inventory Severity Index; BPI-II: Brief Pain Inventory Interference Index; POMA: Tinetti Performance-Oriented Mobility Assessment; FSS: Fatigue Severity Scale; FIM: Functional Independence Measure^.^* *p* ≤ 0.05

## Data Availability

Data will be provided upon reasonable request.

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
