# Peer review of "Pain and Motor Function in Myotonic Dystrophy Type 1: A Cross-Sectional Study"

_ijerph, 2023, doi:10.3390/ijerph20075244_

Round 1
Reviewer 1 Report
The article titled “Pain and motor function in myotonic dystrophy type 1: a cross- 2 sectional study” by Liguori et al. provides further characterization of pain in often overlooked population of DM1 patients.
The authors claim multidimensional assessment and gender differences are not well studied in this population and attempt to shed light on this topic.
Overall it’s a well written manuscript, below are my comments.
1. A recently published paper by Solbakken et al. goes at length to describe pain, function and gender differences in DM1 patients. While the authors refer to this paper at some places, the introduction must describe why the current study complements the existing study by Solbakken. It is not very clear what more the current study adds to existing findings from reading the introduction.
2. Statistical testing seems to be less rigorous. The authors state “Normality was checked through the Shapiro- Wilk test.” What did they find? Most of these datasets are non-normally distributed. If that is the case, authros should report median,IQR rather than mean,SD.
3. Again, the authors state : Wilcoxon rank-sum (Mann–Whitney) test and Chi-square exact test was used when appropriated. But, it appears they only used Wilcoxon.
4. Similarly, the authors say Kruskal-Wallis one-way analysis-of-variance-by-ranks test was used for the factoral design. However, the tables corresponding mentioned they were done by Wilcoxon. It’s very confusion what statistical testing were done for which comparisons.
5. Also, standardized mean differences will help to understand the effect sizes better.
6. How did the authors control for multiple comparisons problem?
7. There are too many tables. Except for table-1 with demographics, the rest of them should be figures (bar plots) so its easy to appreciate differences visualizing them rather than studying the numbers.
8. Figure-1 is not very clear, what do different colors mean?
9. Rather than finding differences between groups, did the authors try to find if there is a correlation between multidimensional assessment and pain outcomes in these patients using regression?
1. The discussion seems to lack a logical flow. It would be better if there are some subheading directing the readers to specific content in the discussion.
Author Response
The article titled “Pain and motor function in myotonic dystrophy type 1: a cross- 2 sectional study” by Liguori et al. provides further characterization of pain in often overlooked population of DM1 patients.
The authors claim multidimensional assessment and gender differences are not well studied in this population and attempt to shed light on this topic.
Overall it’s a well written manuscript, below are my comments.
Dear reviewer,
I would start by thanking you for taking the time to review this manuscript and for adding very valuable and important suggestions for changes to improve understanding and overall transparency of this manuscript. All comments have been carefully addressed, which I hope you find clarity in the revised manuscript
Comments
- A recently published paper by Solbakken et al. goes at length to describe pain, function and gender differences in DM1 patients. While the authors refer to this paper in some places, the introduction must describe why the current study complements the existing study by Solbakken. It is not very clear what more the current study adds to existing findings from reading the introduction.
Thank you for the clarification. Unlike the manuscript of Solbakken et al, our paper aimed to assess the pain intensity and interference in ADLs in DM1 patients using a multidimensional assessment through the BPI and not a unidimensional evaluation through NRS. Moreover, based on the division into 2 groups according to the severity and interference of pain ( BPI-SI and BPI-II ), we investigated the changes in some functional outcomes reported as disabling in this population such as gait and balance (Wiles CM, Busse ME, Sampson CM, Rogers MT, Fenton-May J, van Deursen R. Falls and stumbles in myotonic dystrophy. J Neurol Neurosurg Psychiatry. 2006 Mar;77(3):393-6. doi: 10.1136/jnnp.2005.066258. Epub 2005 Sep 30. PMID: 16199443; PMCID: PMC2077718.; Hammarén E, Kollén L. What Happened with Muscle Force, Dynamic Stability And Falls? A 10-Year Longitudinal Follow-Up in Adults with Myotonic Dystrophy Type 1. J Neuromuscul Dis. 2021;8(6):1007-1016. doi: 10.3233/JND-200521. PMID: 34151851; PMCID: PMC8673550.), not investigated by Solbakken et al. We clarified these points and added the reference in the Introduction, lines 65-67
- Statistical testing seems to be less rigorous. The authors state “Normality was checked through the Shapiro- Wilk test.” What did they find? Most of these datasets are non-normally distributed. If that is the case, authros should report median,IQR rather than mean,SD.
Thank you for the clarification. We modified the paragraph reporting only the statistical tests used and the median and IQR for the data not normally distributed. See lines 134-150
- Again, the authors state : Wilcoxon rank-sum (Mann–Whitney) test and Chi-square exact test was used when appropriated. But, it appears they only used Wilcoxon.
Thank you for the clarification. We modified the paragraph reporting only the statistical tests used. See lines 134-150
- Similarly, the authors say Kruskal-Wallis one-way analysis-of-variance-by-ranks test was used for the factoral design. However, the tables corresponding mentioned they were done by Wilcoxon. It’s very confusion what statistical testing were done for which comparisons.
Thank you for the clarification. We modified the paragraph reporting only the statistical tests used. See lines 134-150
- Also, standardized mean differences will help to understand the effect sizes better.
Thank you for the suggestion. We added the effect size using measures according to the statistical test performed. For the Independent-Samples Mann-Whitney U Test, we reported the effect size (η2) calculated using the formula η2=Z2/N-1. The reference values used indicated a small (0.01), medium (0.06) or large (0.14) effect according to Miles, J and Shevlin, M (2001) Applying Regression and Correlation: A Guide for Students and Researchers. Sage:London. For the Independent-Samples Kruskal-Wallis Test effect size (ε2) reported a 0.00 < 0.01 – negligible, 0.01 < 0.04 – weak, 0.04 < 0.16 – moderate, 0.16 < 0.36 - relatively strong, 0.36 < 0.64 – strong and 0.64 < 1.00 - very strong effect according to Rea, L. M., & Parker, R. A. (1992). Designing and conducting survey research: a comprehensive guide. San Francisco: Jossey-Bass Publishers. See lines 134-150
- How did the authors control for multiple comparisons problem?
Thank you for the suggestion. We performed the Independent-Samples Kruskal-Wallis Test for the comparison and significance values have been adjusted by the Bonferroni correction for multiple tests. We added these data in the text and graphs See lines 146-150
- There are too many tables. Except for table-1 with demographics, the rest of them should be figures (bar plots) so its easy to appreciate differences visualizing them rather than studying the numbers.
Thank you for the suggestion. We modified all the tables (except for table 1) in graphs with legends and abbreviations. We added just another table (n. 2) for the results of the Multivariate Logistic regression analysis.
- Figure-1 is not very clear, what do different colors mean?
Thank you for the suggestion. We modified the figure to improve readability and understanding.
- Rather than finding differences between groups, did the authors try to find if there is a correlation between multidimensional assessment and pain outcomes in these patients using regression?
Thank you for the suggestion. We included the Multivariate Logistic regression analysis to measure the influence of duration disease, FSS, FIM and POMA on pain severity and interference. See Table 2
- The discussion seems to lack a logical flow. It would be better if there are some subheading directing the readers to specific content in the discussion.
Thank you for the suggestion. We modified it accordingly. See lines 439-639
Reviewer 2 Report
This work aims to study pain in a cohort of DM1 patients (23 males and 22 females), emphasizing gender differences. The authors assessed pain severity, interference with daily life activities, gait and balance, functional independence, and fatigue by applying several clinical tests and scales. The manuscript is well-written and provides relevant data on pain perception in DM1 patients.
Comments
1.-To clarify the clinical variability of DM1, could the authors mention in the Introduction section the clinical phenotypes, including all forms of DM1?
2.-Pain is significantly correlated with CTG size (according to reference 11), but the pain has no negative effects on CTG expansion; please correct this assumption (lines 41 to 43).
3.-Could the authors give information about the CTG tract size of the recruited patients? Do patients belong to the classic DM1 phenotype?
4.-Some information is missing in Figure 1. Please explain the color bars (green, severity index SI; blue, interference index II). Please include the y-axis legend, and explain the meaning of the numbers on each bar (frequency).
5.-It would be interesting to know if disease duration (disease progression) affects the perception of pain in this cohort. Is it possible to include this analysis?
Author Response
This work aims to study pain in a cohort of DM1 patients (23 males and 22 females), emphasizing gender differences. The authors assessed pain severity, interference with daily life activities, gait and balance, functional independence, and fatigue by applying several clinical tests and scales. The manuscript is well-written and provides relevant data on pain perception in DM1 patients.
Dear reviewer,
I would start by thanking you for taking the time to review this manuscript and for adding valuable and important suggestions for changes to improve understanding and overall transparency. All comments have been carefully addressed, which I hope you find clarity in the revised manuscript
Comments
1.-To clarify the clinical variability of DM1, could the authors mention in the Introduction section the clinical phenotypes, including all forms of DM1?
Thank you for the suggestion. We modified it accordingly. See lines 40-43
2.-Pain is significantly correlated with CTG size (according to reference 11), but the pain has no negative effects on CTG expansion; please correct this assumption (lines 41 to 43).
Thank you for the suggestion. We clarified accordingly. See lines 45-50
3.-Could the authors give information about the CTG tract size of the recruited patients? Do patients belong to the classic DM1 phenotype?
Thank you for the suggestion. All the patients reported a CTG-repeat size of 50-1000 so we added this information in the methods section. See line 74
4.-Some information is missing in Figure 1. Please explain the color bars (green, severity index SI; blue, interference index II). Please include the y-axis legend, and explain the meaning of the numbers on each bar (frequency).
Thank you for the suggestion. We modified the figure to improve readability and understanding.
5.-It would be interesting to know if disease duration (disease progression) affects the perception of pain in this cohort. Is it possible to include this analysis?
Thank you for the suggestion. According to reviewer 1, we included the Multivariate Logistic regression analysis to measure the influence of duration disease, FSS, FIM and POMA on pain severity and interference. See Table 2